# Irreducible Loss Floors in Gradient Descent Convergence and Energy Footprint

`author@institution.com`

## Abstract

Despite their central role, convergence analyses of the dynamics of loss functions during training require strong assumptions (e.g convexity and smoothness) which are non-trivial to prove. In this work, we introduce a framework for deriving necessary convergence conditions that hold without restrictive assumptions on the dataset or the model architecture. By linking microscopic properties such as individual sample losses and their gradient to macroscopic training dynamics, we derive tight lower bounds for loss functions, applicable to both full-batch and mini-batch gradient systems. These bounds reveal the presence of *irreducible floors* that optimizers cannot surpass and beyond theoretical guarantees, this framework offers a practical tool for anticipating convergence speed, and estimating *minimum training time and energy* requirements. Thus, this framework can be used to ensure the sustainability and feasibility of large-scale training regimes.

## 1 Introduction

Machine learning algorithms are trained using vast amounts of data and backpropagation, which relies on gradient descent. In this process, the loss function is typically defined as the mean of individual samples losses over a training dataset. Thus, the loss function can be seen as a macroscopic observable that reflects the average behavior of individual sample losses. To better understand the dynamics and properties of the loss function, it is essential to analyze the contributions and interactions of these individual local losses. In this research we demonstrate that for a model $x \mapsto f(x, \theta)$ and a dataset $D$, the rate of change of a loss function $\mathcal{L} = \mathcal{L}(\theta, D)$, with respect to latent time $t$ (see. Eq.5) verifies

$$\frac{d\mathcal{L}}{dt} + \mathcal{K}(f, \theta, D)^2 \cdot g(\mathcal{L})^2 \geq 0 \tag{1}$$

with $\theta$ as the model weights and $\mathcal{K}(f, \theta, D)$ the instantaneous decay rate (see Section 4) defined as the sensitivity of the model weights over the dataset $D$, while $x \mapsto g(x)$ is a shape function that defines the structure of the trajectory of $\mathcal{L}$ (see Table 1 and demos at https://github.com/wkzng/iredfloor).

Table 1: Shape functions governing the trajectory of training losses

| | |
|---|---|
| Mean Absolute Error (Section 4.1) | $g(x) = 1$ |
| Mean Squared Error (Section 4.1) | $g(x) = \sqrt{2x}$ |
| Categorical Cross-Entropy (Section 4.2) | $g(x) = \sqrt{2(1 - e^{-x})^2 + 2^{-m+1} \cdot x^2 \cdot e^{-2x}} \, , m \geq 2$ |
| Binary Cross-Entropy (Section 4.3) | $g(x) = \sqrt{\mu x} \quad \text{with} \quad \mu \approx 0.407264$ |

## 2 Related Work

Multiple works in optimization theory applied to Machine Learning studied the behavior of loss function and the speed of convergence for gradient descent learning algorithms in the limit of large training time. Various lines of work explore conditions under which gradient descent exhibits predictable convergence. For instance,Arora et al. [2019] show that under suitable assumptions on hidden dimensionality, weight initialization and initial loss, some architectures converge at a linear rate when trained with the $L_2$ loss. Similarly, Ahmadova [2023] analyze convergence for both gradient descent and gradient flow systems with minimal requirements for the learning rate. Cheridito et al. [2022] also study convergence for gradient descent systems when the target function is constant, providing insights into the convergence in the most simplified settings.

Others works focus on the connection between convergence rate and the geometry of loss landscape. Frei and Gu [2022] construct a unified framework based on proxy-convexity, extending the Polyak-Łojasiewicz (PL) inequality to settings where strong convexity may not be trivial. The PL inequality provides a lower bound for the squared gradient norm, which, when combined the differential equation describing the time evolution of the loss (see. Section 3), enables to derive numerical or analytical upper bounds for the loss and sufficient conditions for convergence. However, since the PL assumptions cannot be trivially proved for all models, Frei and Gu [2022]'s framework provide a more flexible alternative. Similarly, Karimi et al. [2020] derive explicit rates of convergence for gradient and proximal methods with the PL inequality, including stochastic variants (extended in Section 6). In contrast to these works, our approach does not assume PL-like conditions, leading instead to necessary (rather than sufficient) criteria for convergence.

Other lines of work focuse in empirical observations of loss dynamics in large language models (LLM). Luo et al. [2025] study multi-power laws (MPL) for language model pretraining loss curves which capture the effect of the learning rate schedule (revisited by Eq. 5). In addition, Maloney et al. [2022] (Sections 3 and 4) analyze properties which allow the emergence of neural scaling laws Kaplan et al. [2020] for language models. These studies underscore the role of both optimization dynamics and architectural factors in shaping the dynamics behavior of gradient descent systems.

## 3 Differential Equation for Loss Functions

We now recall the key ideas used to derive irreducible floors for loss functions from first principles. To that end, consider a training set $D$, which consists of $n$ samples $D = \{(x_i, y_i) \mid i = 1, \ldots, n\}$ where each $x_i$ is an input sample, and $y_i$ is its corresponding target. In the unsupervised learning case, the $y_i$ are omitted. The average loss over the dataset is herein defined as

$$\mathcal{L}(\theta, D) = \mathbb{E}\big[\ell\big] = \sum_{i=1}^{n} \ell_i \cdot \mathbb{P}(\ell = \ell_i) \tag{2}$$

where $\ell_i = \ell_i(y_i, f(x_i, \theta))$ is the pointwise loss which measures the discrepancy between a model output and its target (or a suitable proxy in unsupervised learning). Without loss of generality, we can consider the data sampling $\mathbb{P}(\ell)$ to be uniform over the dataset i.e $\mathbb{P}(\ell = \ell_i) = 1/n$. Thus training a model requires to minimize $\mathcal{L}(\theta, D)$, often through gradient-based optimization. In the continuous-time limit, gradient descent can be interpreted as the gradient flow dynamics

$$\frac{d\theta}{dt} = -\nabla \mathcal{L}(\theta, D) \tag{3}$$

This differential equation describes the time evolution of the model's weights as they move along the negative gradient of the loss function to minimize it. The dynamics of this process thus depends on the model architecture, the loss landscape and the overall optimization strategy. As the weights move along its gradient, the loss function itself decreases with an instantaneous rate of change given by

$$\frac{d}{dt}\mathcal{L}(\theta, D) = \nabla \mathcal{L}(\theta, D) \cdot \frac{d\theta}{dt} = -\big\|\nabla \mathcal{L}(\theta, D)\big\|^2 \tag{4}$$

The negative sign ensures its decrease over time with a rate of change which depends on the square of its gradient norm. Therefore, analyzing it provides key insights into the learning process. Furthermore, the gradient descent update rule $\theta_k = \theta_{k-1} - lr_k \cdot \nabla\mathcal{L}(\theta_k, D)$ suggests that the latent time of the weights and loss dynamics can be expressed as the integral of the learning rate schedule i.e

$$t_k = t_{k-1} + lr_{k-1} = \sum_{i=0}^{k-1} lr_i \approx \int_0^{k-1} lr(\tau)d\tau \qquad (5)$$

## 4 Upper Bounds for The Loss Gradient Norm

To derive the irreducible loss differential inequality (see Eq. 1), we determine upper bounds for $\|\nabla\mathcal{L}\|$ as functions that depend explicitly on $\mathcal{L}$. To achieve tight upper bounds, ours computations are guided by the standard ordering of $L_p$ norms.[1] Though we derive these upper bounds for single-component losses and without regularization, Section 5 extends these results to regularized and multi-component losses.

### 4.1 Mean Absolute Error and Mean Squared Error losses

Here, we derive an upper bound for the gradient of the MAE (or $L_1$) loss and the MSE (or $L_2$) loss. To that end, we start from the pointwise loss $\ell = \frac{1}{\alpha}|y - f(x,\theta)|^\alpha$ with $\alpha \in \{1, 2\}$. It follows that

$$\nabla\ell = -\mathbf{sign}(y - f(x,\theta)) \cdot g(\alpha\ell) \cdot \nabla f \qquad (6)$$

With $g(x) = x^{-\frac{1}{\alpha}+1}$. Therefore, by applying the gradient operator on Eq. 2, followed by the triangle and the Cauchy-Schwarz inequalities, we obtain

$$\left\|\frac{1}{n}\sum_{i=1}^n \nabla\ell_i\right\| \leq \frac{1}{n}\sum_{i=1}^n \|\nabla\ell_i\| \leq \sqrt{\frac{1}{n}\sum_{i=1}^n g(\alpha\ell_i)^2} \cdot \sqrt{\frac{1}{n}\sum_{i=1}^n \|\nabla f(x_i,\theta)\|^2} \qquad (7)$$

Furthermore, $x \mapsto g(x)^2$ is concave for $1 \leq \alpha \leq 2$ where the Jensen inequality yields

$$\|\nabla\mathcal{L}\| \leq \mathcal{K} \cdot (\alpha\mathcal{L})^{-\frac{1}{\alpha}+1} \quad \text{with} \quad \mathcal{K} = \sqrt{\frac{1}{n}\sum_{i=1}^n \|\nabla f(x_i,\theta)\|^2} \qquad (8)$$

For the MAE loss ($\alpha = 1$), the upper bound for $\|\nabla\mathcal{L}_{\text{MAE}}\|$ remains independent of $\mathcal{L}_{\text{MAE}}$. This is a well-known behavior of the $L_1$ loss which can have non-zero gradient even when the loss itself is zero. Note that a tighter value for $\mathcal{K}$ can be obtained directly with the triangle inequality since $g(x) = 1$. In contrast, the MSE loss ($\alpha = 2$) has an upper bound for $\|\nabla\mathcal{L}_{\text{MSE}}\|$ which depends on $\sqrt{\mathcal{L}_{\text{MSE}}}$ whose zero ensures the optimality of the model over the training data which occurs when the model has perfectly fitted all the training samples.

### 4.2 Multi-Class Categorical Cross-Entropy

We analyze the loss for a multi-class classification problem where the number of output classes is denoted as $C$. The pointwise loss is given by

$$\ell = -\sum_{k=1}^C y_k \log p_k \quad \text{with} \quad p_k = \mathbf{softmax}(z) = \frac{e^{z_k}}{\sum_{i=1}^C e^{z_i}} \qquad (9)$$

where $z = f(x,\theta)$ is the logits vector. A well-establish result is $\frac{\partial\ell}{\partial z_k} = p_k - y_k$ (can be shown with the chain-rule). Thus, applying the triangle then the Cauchy-Schwarz inequalities yields

---

[1]Distances in $\mathbb{R}^d$ can be measured with multiple norms which generalize the well-established euclidean norm as follows: $\frac{1}{d}||x||_1 \leq \frac{1}{\sqrt{d}}||x||_2 \leq ||x||_\infty \leq ||x||_2 \leq ||x||_1$ where $x \in \mathbb{R}^d$

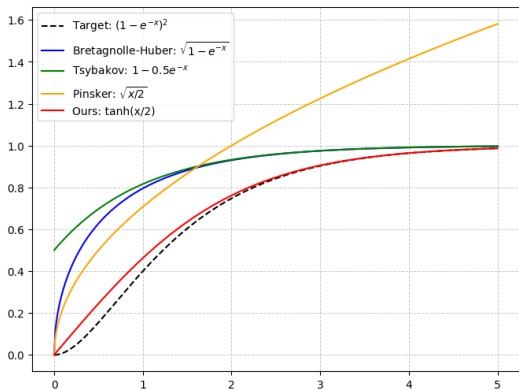

Figure 1: Comparison of uniformly concave upper-bounds for $x \mapsto (1 - e^{-x})^2$. The Pinsker, Bretagnolle-Huber and Tsybakov functions Canonne [2023] are often used to study total variation of probability measures defined for two probability measures P and Q as $d_{TV}(P, Q) = \sum_k |P_k - Q_k|$

$$\|\nabla\ell\| = \left\| \sum_{k=1}^{C} \frac{\partial\ell}{\partial z_k} \nabla z_k \right\| \leq \sqrt{\sum_{k=1}^{C} |p_k - y_k|^2} \cdot \sqrt{\sum_{k=1}^{C} \|\nabla z_k\|^2} \tag{10}$$

For multi-class classification, only one of the $y_k$ is equal to 1 while the remainders are all equal to zero. Let us denote its index as $m$. This leads to $\ell = -\log(p_m)$, meaning that $p_m = e^{-\ell}$ and

$$\sum_{k=1}^{C} |p_k - y_k|^2 = \left( \sum_{k \neq m}^{C} p_k^2 \right) + (1 - p_m)^2 \leq \left( \sum_{k \neq m}^{C} p_k \right)^2 + (1 - p_m)^2 = 2 \cdot (1 - p_m)^2 \tag{11}$$

Therefore any pointwise loss verifies $\|\nabla\ell\| \leq \sqrt{2} \cdot \left(1 - e^{-\ell}\right) \cdot \|\nabla f\|$, recalling that $z = f(x, \theta)$. Applying the triangle and Cauchy-Schwarz inequalities again, we obtain

$$\left\| \frac{1}{n} \sum_{i=1}^{n} \nabla\ell_i \right\| \leq \frac{1}{n} \sum_{i=1}^{n} \|\nabla\ell_i\| \leq \sqrt{\frac{2}{n} \sum_{i=1}^{n} \left(1 - e^{-\ell_i}\right)^2} \cdot \sqrt{\frac{1}{n} \sum_{i=1}^{n} \|\nabla f(x_i, \theta)\|^2} \tag{12}$$

For the subsequent analysis, we introduce $g : x \mapsto 1 - e^{-x}$, which is increasing, uniformly concave and maps $[0, +\infty)$ to $[0, 1)$. This leads to reframing the leftmost term of the right-hand side (RHS) as

$$\frac{1}{n} \sum_{i=1}^{n} \left(1 - e^{-\ell_i}\right)^2 = \mathbb{E}\left[\left(1 - e^{-\ell}\right)^2\right] = \mathbb{E}[g(\ell)^2] \tag{13}$$

The Jensen inequality would produce a suitable upper bound for $\mathbb{E}[g(\ell)^2]$ as long as $x \mapsto g(x)^2$ is concave. However, it is only the case when $x > \log 2 \approx 0.693$ but since $\ell_i > \log 2$ is not always verified, we explored uniformly concave upper bounds. Drawing inspiration from Canonne [2023], we found $x \mapsto \tanh(x/2) = \frac{1-e^{-x}}{1+e^{-x}}$ to be an adequate candidate (see Figure 1) that leads to

$$\mathbb{E}\left[g(\ell)^2\right] \leq \mathbb{E}\left[\tanh(\ell/2)\right] \leq \tanh\left(\mathbb{E}[\ell]/2\right) = \tanh(\mathcal{L}/2) \tag{14}$$

Although a close approximation, $x \mapsto \tanh(x/2)$ does not capture the curvature of $x \mapsto g(x)^2$ when $x$ is small. To account for this curvature, we use the uniform concavity of $g$ to establish $g(x) \leq g(\mathbb{E}[\ell]) + g'(\mathbb{E}[\ell]) \cdot (x - \mathbb{E}[\ell])$ with $\mathbb{E}[\ell] = \mathcal{L}$. Thus, taking the square and expectation yields

$$\mathbb{E}\left[g(\ell)^2\right] \leq \left(1 - e^{-\mathcal{L}}\right)^2 + \text{Var}(\ell) \cdot e^{-2\mathcal{L}} \tag{15}$$

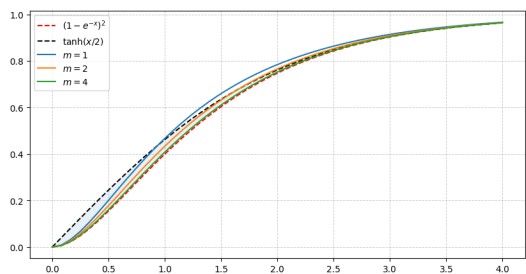

Figure 2: Values for $m$ such that $\left(1 - e^{-x}\right)^2 + 2^{-m} \cdot x^2 \cdot e^{-2x}$ remains close to $\tanh(x/2)$.

Without further information on the exact distribution of the pointwise losses, a simple upper bound cannot be fully determined. Nevertheless, Eq.14 provides a guiding upper bound to analyse admissible expressions for $\mathrm{Var}(\ell)$ to keep the RHS of Eq.15 in the vicinity of $\tanh(\mathcal{L}/2)$, less than 1 and and dominated by $\mathcal{L}^2$. The exploration of such candidates is guided by the Bhatia and Davis [2000] inequality which states that $\mathrm{Var}(\ell) \leq (\mathbb{E}[\ell] - \ell_{min})(\ell_{max} - \mathbb{E}[\ell])$. In particular $\ell_{min} = 0$ and since $\ell_{max}$ and $\mathbb{E}[\ell]$ are two positive real numbers, there exists peak-to-mean rates $1 \leq \alpha \leq n$ such that $\ell_{max} \leq \alpha\mathbb{E}[\ell]$, which implies that $\mathrm{Var}(\ell) \leq (\alpha - 1)\mathcal{L}^2$. In order to keep the RHS of Eq.15 in the vicinity of $\tanh(\mathcal{L}/2)$, we found $\mathrm{Var}(\ell) \leq 2^{-m}\mathcal{L}^2$ with $m \geq 2$ to be suitable candidates (see Figure 2). As a consequence,

$$\|\nabla\mathcal{L}\| \leq \mathcal{K}\sqrt{2\left(1 - e^{-\mathcal{L}}\right)^2 + 2^{-m+1} \cdot \mathcal{L}^2 \cdot e^{-2\mathcal{L}}} \quad \text{with} \quad \mathcal{K} = \sqrt{\frac{1}{n}\sum_{i=1}^{n}\left\|\nabla f(x_i, \theta)\right\|^2} \quad (16)$$

As in Section 4.1, this result shows the emergence of an upper-bound for $\|\nabla\mathcal{L}_{\mathrm{CCE}}\|$ which explicitly depends on $\mathcal{L}_{\mathrm{CCE}}$. Furthermore, bounds for variance verifying $\mathrm{Var}(\ell) = \mathcal{O}(\mathbb{E}[\ell]^2)$ are typical for light-tailed distributions, meaning that this upper bound captures a coordinated decrease of pointwise losses with limited marginal sample losses keeping high values compared to the mean.

### 4.3 Multi-Label Binary Cross-Entropy

We herein analyze the loss of a multi-label classification problem where the number of output classes is denoted as $C$. The pointwise loss is given by

$$\ell = -\sum_{j=1}^{C} y_k \log p_k + (1 - y_k)\log(1 - y_k) \quad \text{with} \quad p_k = \textbf{sigmoid}(z_k) = \frac{1}{1 + e^{-z_k}} \quad (17)$$

with $z = f(x, \theta)$. As in Section 4.2, we have $\frac{\partial \ell}{\partial z_k} = p_k - y_k$ and with the same strategy, we obtain

$$\|\nabla\ell\| \leq \|\nabla f\| \cdot \sqrt{\sum_{k=1}^{C} |p_k - y_k|^2} \quad (18)$$

Since $y_k \in \{0, 1\}$ and $p_k \in (0, 1)$, we can distinguish two cases summarized in Table 2

Table 2: Binary cases to cover in order to determine the dynamics shape function

| Case | $|p_k - y_k|^2$ | $y_k \log p_k + (1 - y_k)\log(1 - p_k)$ |
|---|---|---|
| $y_k = 0$ | $p_k^2$ | $\log(1 - p_k)$ |
| $y_k = 1$ | $(1 - p_k)^2$ | $\log(p_k)$ |

Thus by studying the variations of $x \mapsto (1 - x)^2 + \mu\log(x)$ with $x \in (0, 1)$ and $\mu \approx 0.407264$ we obtain $(1 - x)^2 \leq -\mu\log(x)$ and by symmetry $x^2 \leq -\mu\log(1 - x)$. Figure 3 illustrates the key intuition. Consequently we get $|y_k - p_k|^2 \leq -\mu\Big(y_k \log p_k + (1 - y_k)\log(1 - p_k)\Big)$ and

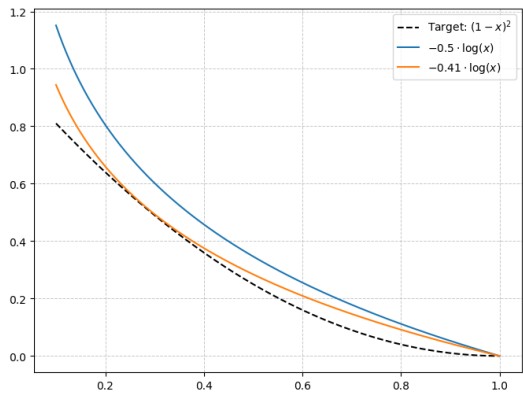

Figure 3: Convex upper-bounds for $x \mapsto (1-x)^2$ in $(0, 1)$. Parametric functions $\phi(x, \mu) = -\mu \log(x)$ provide suitable approximations with an optimal value $\mu \approx 0.407264$ obtained numerically.

$$\sum_{k=1}^{C} |p_k - y_k|^2 \leq \mu \left( -\sum_{k=1}^{C} y_k \log p_k + (1 - y_k) \log(1 - y_k) \right) = \mu \cdot \ell \tag{19}$$

As in Section 4.1 with the Cauchy-Schwarz inequality, we obtain an upper bound depending on the average loss itself in a form that resembles the upper bound obtained for the mean squared error loss.

$$\|\nabla \mathcal{L}_{\text{BCE}}\| \leq \mathcal{K} \sqrt{\mu \mathcal{L}_{\text{BCE}}} \quad \text{with} \quad \mathcal{K} = \sqrt{\frac{1}{n} \sum_{i=1}^{n} \|\nabla f(x_i, \theta)\|^2} \tag{20}$$

This result also bears similarities with the Pinsker function (see. Canonne [2023] and Figure 1) for which $\mu = 0.5$. Our optimal value for the parameter $\mu$ being less than 0.5 implies that we get a tighter upper bound.

## 5 Extension to Multi-Component Loss and Regularization

In the previous section, we deliberately omitted regularization to focus on core convergence dynamics. However, in practice, training often involves an additional regularization term that encourages simpler models, typically by penalizing large weight magnitudes. Moreover, a training loss is often made up of multiple independent components (e.g. VAE Kingma and Welling [2013], VQ-VAE van den Oord et al. [2018]). A regularized loss with $p \geq 1$ components $\mathcal{L}_i \geq 0$ for $i = 1, \ldots, p$ is formulated as

$$\mathcal{L}_*(\theta, D) = \sum_{i=1}^{p} \omega_i \cdot \mathcal{L}_i(\theta, D) + \lambda \cdot \mathcal{R}(\theta) \tag{21}$$

where $\omega_i > 0$ is the weight of the i-th component, $\mathcal{R}(\theta)$ is a regularization function (e.g., $L_2$ norm), and $\lambda > 0$ controls its influence. This modification affects both the optimization trajectory and the ultimate convergence floor, and must therefore be incorporated into any realistic analysis of training dynamics. In most settings, $\mathcal{R}(\theta) = \frac{1}{2}\|\theta\|^2$ whose gradient verifies $\|\nabla \mathcal{R}\| = \sqrt{2\mathcal{R}}$. Following the results established in Section 4, the loss components have gradients with bounded norm of the form $\|\nabla \mathcal{L}_i\| \leq \mathcal{K} \cdot g_i(\mathcal{L}_i)$ with $g_i$ as positive and increasing shape function. Moreover, $\lambda \mathcal{R} \leq \mathcal{L}_*$ and any component of the training loss verifies $\omega_i \mathcal{L}_i \leq \mathcal{L}_*$. Thus applying the triangle inequality yields

$$\|\nabla \mathcal{L}_*\| \leq \mathcal{K} \cdot \left( \sum_{i=1}^{p} \omega_i \cdot g_i(\mathcal{L}_*/\omega_i) \right) + \sqrt{2\lambda \mathcal{L}_*} \leq (\mathcal{K} + \sqrt{\lambda}) \cdot \left( \sum_{i=1}^{p} \omega_i \cdot g_i(\mathcal{L}_*/\omega_i) + \sqrt{2\mathcal{L}_*} \right) \tag{22}$$

Consequently, the gradient of a regularized multi-component loss remains bounded by a function that depends on the loss itself. In particular, the added $L_2$ regularization term contributes a constant factor effectively acting as a stable offset in the decay rate.

## 6   Extension to Stochastic Gradient Flow

The dynamics equations in Section 3 are established when the entire dataset $D$ is used at each time step, which may be unpractical for large models. In practice, most models are trained using mini-batch with batch size $m \leq |D|$. Thus, the loss over a batch $B(t)$ is expressed with $n = |D|$ as

$$\mathcal{L}(\theta, B(t)) = \frac{1}{m} \sum_{i=1}^{n} q_i(t) \, \ell_i \tag{23}$$

where $q_i(t) = 1$ if sample $i$ is included in $B(t)$ and 0 otherwise. When training on a large dataset, mini-batch selection across multiple epochs can be modeled as a uniform sampling without replacement within each epoch but with replacement across epochs. Let $q(t) = [q_1(t), \ldots, q_n(t)]$ be the binary indicator vector for the batch at time $t$. There are exactly $\binom{n}{m}$ possible batches of size $m$ and each sample has inclusion probability $\binom{n-1}{m-1}/\binom{n}{m} = m/n$. Consequently, $\mathbb{E}[q_i(t)] = m/n$ and

$$\mathbb{E}\big[\mathcal{L}(\theta, B(t))\big] = \frac{1}{m} \sum_{i=1}^{n} \mathbb{E}\big[q_i(t)\big] \cdot \ell_i = \mathcal{L}(\theta, D) \tag{24}$$

This result showcases that despite the stochastic nature of mini-batch flow, the batch loss is on expectation the same as the average loss over the entire dataset. Following a similar strategy (see complete proofs in Appendix A.1), we can show that $\mathrm{Var}[q_i(t)] = \frac{m}{n}\left(1 - \frac{m}{n}\right)$ and $\mathrm{Cov}(q_i(t), q_j(t)) = -\frac{m(n-m)}{n^2(n-1)}$ for $i \neq j$, leading to

$$\mathrm{Var}\Big[\mathcal{L}(\theta, B(t))\Big] = \frac{n-m}{n-1} \cdot \frac{\mathrm{Var}(\ell)}{m} \tag{25}$$

The variance of $\mathcal{L}(\theta, B(t))$ is a fraction of the intrinsic variance of the sample losses. Thus applying the *Bienaymé-Tchebychev* inequality yields a concentration bound which implies that the mini-batch loss remains close to the full-dataset loss with high probability, provided that the batch size is sufficiently large. As a result, the batch size $m$ can be selected based on the desired deviation tolerance $\mathbb{P}\Big(\big|\mathcal{L}(\theta, B(t)) - \mathcal{L}(\theta, D)\big| \geq \alpha \cdot \mathrm{Var}(\ell)^{1/2}\Big) \leq \frac{n-m}{m(n-1)\alpha^2} \leq \frac{1}{m\alpha^2}$ where $\alpha > 0$ is the deviation parameter. Figure 4 shows how large batch sizes provide close estimates of the dataset loss.

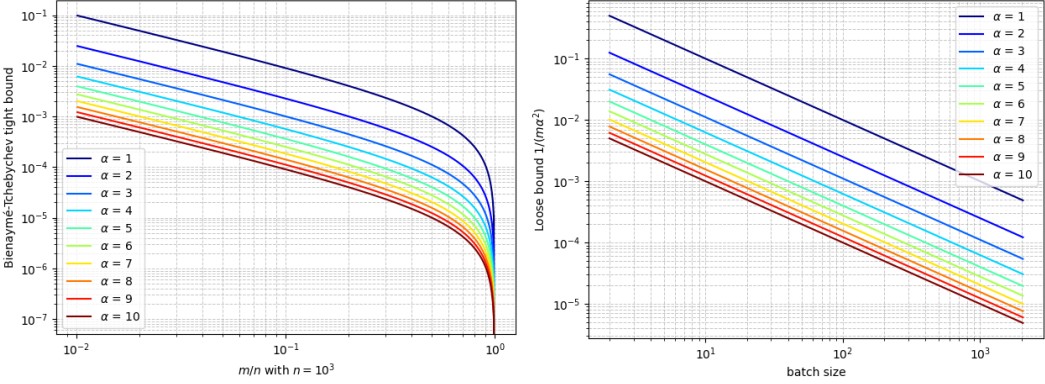

Figure 4: The tight Bienaymé–Tchebychev bound (left) decreases faster than the loose but universal bound (right) due to the non-linear dependency on the dataset size and batch size. However both dynamics shows that the bound always becomes tighter as either the batch size or the deviation parameter $\alpha$ increases.

# 7 Necessary Convergence Condition and Energy Footprint

A key claim of our research is that irreducible floors are closely related to energy footprint. To establish that link, we start by studying the link between the emergence of irreducible loss floors and necessary convergence conditions, first illustrated on a simple linear model $f(x, \theta) = \theta^T x$ whose gradient is $\nabla f(x, \theta) = x$. From this it follows that $\mathcal{K} = \sqrt{\frac{1}{n} \sum_{i=1}^{n} \|x_i\|^2}$ where $n$ is the dataset size. For this model, the loss decay rate is independent of the model weights (and thus also time-independent). This simplifies the resolution of Eq. 1 for various losses as shown in Table 3

Table 3: Analytical solutions for the loss dynamics

| Loss Function | Irreducible Floor Inequality | Analytical Solution |
|---|---|---|
| Mean Absolute Error | $\frac{d\mathcal{L}}{dt} + \mathcal{K}^2 \geq 0$ | $\mathcal{L} \geq \mathcal{L}_0 - \mathcal{K}^2 t$ |
| Mean Squared Error | $\frac{d\mathcal{L}}{dt} + 2\mathcal{K}^2 \mathcal{L} \geq 0$ | $\mathcal{L} \geq \mathcal{L}_0 e^{-2\mathcal{K}^2 t}$ |
| CCE (with $m \to \infty$) | $\frac{d\mathcal{L}}{dt} + 2\mathcal{K}^2 (1 - e^{-\mathcal{L}})^2 \geq 0$ | $\log\left(\frac{e^{\mathcal{L}} - 1}{e^{\mathcal{L}_0} - 1}\right) + \frac{e^{-\mathcal{L}_0} - e^{-\mathcal{L}}}{(1 - e^{-\mathcal{L}_0})(1 - e^{-\mathcal{L}})} \geq -2\mathcal{K}^2 t$ |

In the general case, whether through analytical derivation or numerical simulation, we can estimate the minimum latent time required for the loss to fall below a given threshold $\epsilon$ as a function $\Phi(\epsilon, \mathcal{K}, \mathcal{L}_0)$ linking the convergence time to the initial loss, the model architecture and the dataset. For instance, the MSE loss on a linear model implies that the required latent time for $\mathcal{L}(t) \leq \epsilon$ verifies

$$t \geq \frac{1}{2\mathcal{K}^2} \log\left(\frac{\mathcal{L}_0}{\epsilon}\right) = \Phi(\epsilon, \mathcal{K}, \mathcal{L}_0) \tag{26}$$

Furthermore, assuming a constant learning rate reduces Eq. 5 to $t_k = lr \cdot k$. To relate latent time to actual training duration, we introduce two quantities : $C$ as the compute rate defined as the number of FLOPs executed per unit of real time and $S(f, B(t))$ as the number of FLOPs required for one forward-backward pass on a mini-batch. Under these assumptions, the actual training time required to for the training loss to fall below $\epsilon$ satisfies the inequality

$$T = \text{training duration} \geq \frac{S(f, B(t))}{C} \cdot \frac{\Phi(\epsilon, \mathcal{K}, \mathcal{L}_0)}{lr} \tag{27}$$

Next, assuming that the power consumption $P$ is constant during training (or with an equivalent measure of the average consumption), we can estimate the minimum energy footprint as

$$E = P \cdot \frac{S(f, B(t))}{C} \cdot \frac{\Phi(\epsilon, \mathcal{K}, \mathcal{L}_0)}{lr} \tag{28}$$

In practice, the power consumed is closely linked to the number of FLOPs and the memory footprint of both the model and dataset. We conjecture that $P$ can be approximated as a function of the model, dataset, batch size and compute rate i.e $P = \mathcal{P}(f, D, B(t), C)$. Such relation enables apriori estimation of the total energy required to train a given model to a target loss threshold and opens pathways for sustainable machine learning, where energy efficiency and carbon footprint could be forecasted and optimized (e.g model architecture engineering) before scaling the training process.

# 8 Exploring a Practical Limit of the Framework

In this section, we explore a current practical limitation of the irreducible floor framework by applying it to the MNIST LeCun and Cortes [2010] classification task. Specifically, we compare the empirical training loss with numerically computed irreducible floors using an ODE solver. All experiments were conducted on Google Colab using a T4-GPU runtime and PyTorch version 2.6.0+cu124. Due to computational constraints, we used a compact MLP with 0.8M parameter as well as a compact

CNN with 1.8M parameters both trained with a batch size of 128 and a fixed learning rate $10^{-4}$. The implementation, available at https://github.com/wkzng/iredfloor, is adaptable to larger models but was tested here under restricted compute conditions.

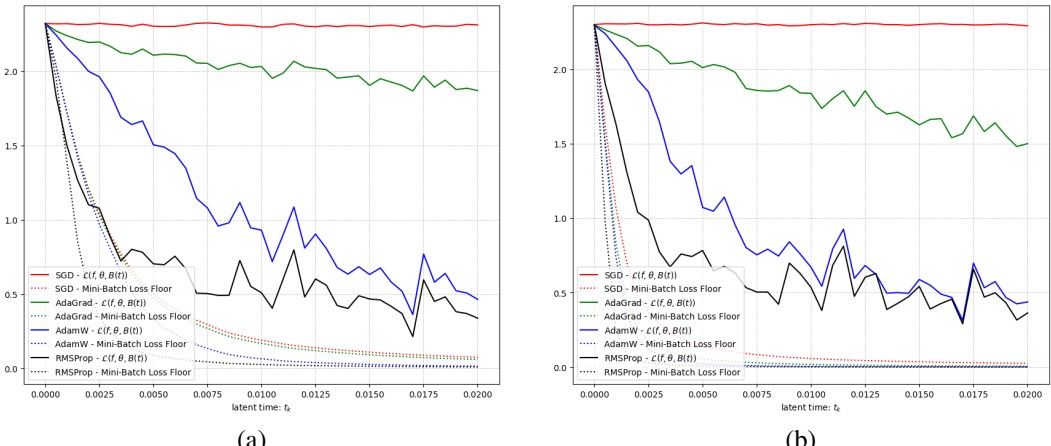

(a)            (b)

Figure 5: Time evolution of an MLP (a) and CNN (b) training losses for various optimizers (SGD, AdaGrad, AdamW, RMSProp) on the MNIST dataset for 100 back-propagation steps.

As illustrated in Figure 5, the irreducible floor for SGD is never crossed. However, alternative optimizers such as RMSProp may fall below their respective floor. This can be attributed to the fact that we did not use the entire dataset at each step (see probabilistic bounds in Section 6), as well as the fact that these advanced optimizers do not strictly follow the classical gradient descent update rule. The inclusion of momentum terms and adaptive learning rate scaling modifies the dynamics, transforming the gradient flow described by Eq. 3 into a differential equation of the form

$$\frac{d\theta}{dt} = -\omega\left(\theta, \nabla\mathcal{L}\right) \cdot \nabla\mathcal{L} \tag{29}$$

where $\omega(.)$ denotes an operator that incorporates additional dynamics specific to each optimizer, such as accumulated gradients or moving averages. This altered flow implies that Eq. 1 must be modified to account for these additional dynamics, leading to the generalized differential inequality

$$\frac{d\mathcal{L}}{dt} + \Omega\left(\mathcal{L}\right) \cdot \mathcal{K}(f, \theta, D)^2 \cdot g(\mathcal{L})^2 \geq 0 \tag{30}$$

in which $\Omega$ is a functional of the training loss that serves as a tight upper bound for $|\omega|$, that depends only on the instantaneous value of the training loss. This abstraction allows the convergence analysis to be extended to optimizers that deviate from simple gradient descent.

## 9 Conclusion and Outlook

In this research, we presented a general framework for analyzing necessary convergence conditions of gradient-based optimization algorithms, without relying on strong assumptions such as convexity or smoothness. Additionally, we introduced a practical method to estimate the minimal energy required to reach a target performance, thereby providing keys to support the development of more sustainable and compute-efficient training pipelines.

That said, the framework remains in an early stage and requires several enhancements for practical deployment with modern optimizers and advanced training techniques aimed at improving stability and convergence. Notably, estimating loss decay rates prior to training remains an open challenge, while in-training estimation are computationally expensive. A refinement of the irreducible floor differential inequality is proposed in Section 8 to better capture optimizer-specific dynamics and yield tighter bounds across a broader class of algorithms. Lastly, extending the analysis in Section 6 to estimate irreducible floors on unseen test data remains an important direction for future work.

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

## Appendix : Additional Technical Proofs

### A.1 Variance of the Stochastic Gradient Flow Loss

In Section 6 we established that $q_i(t) \in \{0, 1\}$ is a binomial random variable with realization probability $p = \mathbb{P}(q_i(t) = 1) = m/n$. Using known results with binomial random variable, we obtain $\mathbb{E}[q_i(t)] = p$ and $\text{Var}[q_i(t)] = p \cdot (1 - p) = \frac{m(n-m)}{n^2}$. Furthermore if $j \neq i$, we have $\mathbb{P}(q_i(t) = 1, q_j(t) = 1) = \frac{\binom{n-2}{m-2}}{\binom{n}{m}} = \frac{m(m-1)}{n(n-1)}$, which leads to

$$\text{Cov}(q_i(t), q_j(t)) = \frac{m(m-1)}{n(n-1)} - \left(\frac{m}{n}\right)^2 = -\frac{m(n-m)}{n^2(n-1)}.$$

Therefore

$$\text{Var}\left(\mathcal{L}(\theta, B(t))\right) = \text{Var}\left(\frac{1}{m}\sum_{i=1}^{n} q_i(t)\,\ell_i\right) = \frac{1}{m^2}\left(\sum_{i=1}^{n}\text{Var}[q_i]\ell_i^2 - 2\sum_{i<j}^{n}\text{Cov}(q_i(t), q_j(t))\ell_i\ell_j\right)$$

After substituting the expressions for the variances and covariances we obtain the simplified expression

$$\text{Var}\left(\mathcal{L}(\theta, B(t))\right) = \frac{n-m}{m(n-1)}\left(\frac{n-1}{n^2}\sum_{i=1}^{n}\ell_i^2 - \frac{2}{n^2}\sum_{i<j}^{n}\ell_i\ell_j\right)$$

The identification of the term in brackets with $\text{Var}(\ell)$ can be established with

$$\text{Var}(\ell) = \left(\frac{1}{n}\sum_{i=1}^{n}\ell_i^2\right) - \left(\frac{1}{n}\sum_{i=1}^{n}\ell_i\right)^2 = \left(\frac{1}{n}\sum_{i=1}^{n}\ell_i^2\right) - \frac{1}{n^2}\left(\sum_{i=1}^{n}\ell_i^2 + 2\sum_{i<j}^{n}\ell_i\ell_j\right)$$

### A.2 Implicit Analytical Solution for the Cross-Entropy Loss for Linear Models

Applying the separation of variable in the differential inequality of the categorical cross-entropy (Section 7) leads to computing a definite integral of the form

$$\int_a^b \frac{dx}{(1 - e^{-x})^2}$$

by using the substitution

$$u = 1 - e^{-x} \quad \Rightarrow \quad du = e^{-x}dx = (1-u)dx \quad \Rightarrow \quad dx = \frac{du}{1-u}$$

that integral becomes

$$\int_a^b \frac{dx}{(1-e^{-x})^2} = \int_{u(a)}^{u(b)} \frac{1}{u^2} \cdot \frac{du}{1-u} = \int_{u(a)}^{u(b)} \frac{du}{u^2(1-u)}$$

Next, we decompose the integrand using partial fractions:

$$\frac{1}{u^2(1-u)} = \frac{1}{u^2} + \frac{1}{u} + \frac{1}{1-u}$$

Therefore, the integral can be computed as

$$\int_{u(a)}^{u(b)} \left(\frac{1}{u^2} + \frac{1}{u} + \frac{1}{1-u}\right)du = \left[-\frac{1}{u} + \log(u) - \log(1-u)\right]_{u=u(a)}^{u=u(b)}$$

The log terms can be further simplified using $u = 1 - e^{-x}$ as

$$\log(u) - \log(1-u) = \log(1 - e^{-x}) - \log(e^{-x}) = \log(e^x - 1)$$

