# OpenReview forum: "Irreducible Loss Floors in Gradient Descent Convergence and Energy Footprint"
_NeurIPS.cc/2025/Conference — Submitted to NeurIPS 2025_

### Official Review · Reviewer_duBp · 2025-06-24

**Clarity:** 3
**Significance:** 2
**Originality:** 3
**Rating:** 4
**Confidence:** 3

**Summary:**

The paper examines the connection between gradient flow and the instantaneous rate of change for commonly used loss functions in machine learning. Across four studied cases, a structural relationship is identified, which allows for the derivation of a lower bound on the convergence behavior of the loss function. This bound can be used to estimate worst-case training times and to provide worst-case estimates for the energy consumption of the method.

**Questions:**

- Can the approach be extended to additional loss functions?
- How tight are the derived bounds?
- Is it possible to compare them to other lower bounds, such as the convergence rate of $\theta$?

**Ethical Concerns:**

["NO or VERY MINOR ethics concerns only"]

**Final Justification:**

Although I find the proposed approach to establishing lower bounds interessting, the lack of clarity regarding tightness and effectiveness is the reason I’m standing by my current score.

**Limitations:**

Yes.

**Paper Formatting Concerns:**

None.

**Quality:**

3

**Strengths And Weaknesses:**

**Strengths**
- Very well-written and easy-to-understand paper.
- Using simple mathematics, it reveals a connection that wasn't previously known in this way. That's why I think it's a very interesting approach that should be pursued further.
- No assumptions about smoothness or convexity are needed to derive the lower bound on training time.
Seems useful for estimating the effort required to train large models.

**Weaknesses**
- The theory is limited to the standard (stochastic) gradient method; regularization can be incorporated. However, in the toy example presented, the approach does not work for common variations like RMSProp. An extension of the approach is briefly discussed, but its correctness cannot be verified without further details.
- The tightness of the derived bounds is not discussed. It is unclear how, or whether, the concept can be generalized to other loss functions beyond the four considered here

---

> ### Author Rebuttal · Authors · 2025-07-27
>
> Thank you for your thoughtful and encouraging review. We greatly appreciate your positive remarks and the constructive questions you raised. Below, we address each point in turn.
>
> 1. Extending the approach to additional loss functions
>
> Yes, we believe the framework can be extended beyond the four loss functions discussed in the paper. As shown in Section 5, if a new loss is a linear combination of losses for which shape functions are already known (see Table 1), then the composite shape function can be derived via Eq. 22. For entirely new loss functions, one can follow the procedure outlined in Section 4. The key requirement is that the norm of the gradient can be expressed (or as tightly as possible bounded) as a function of the loss itself. We showed this structure holds for widely used objective (MSE, MAE, CCE, and BCE) and we anticipate that similar derivations are feasible for other convex losses and even certain non-convex surrogates. The main technical challenge lies in leveraging the tightest possible inequalities when linking individual losses to their gradients.
>
>
> 2. Tightness of the derived bounds
>
> We agree that the tightness of the proposed bounds merits deeper discussion. As noted in Section 4, our inequalities are derived under minimal assumptions and thus trade tightness for generality. The tightness is arguably linked to the possibility of saturation i.e. transforming $\leq$ symbols into $=$ in all inequalities used (triangle, Cauchy-Schwarz, Jensen, etc...) that we used. A discussion that you did not include and that can provide a clearer answer is to analyze the distance $| || \nabla \mathcal{L} || - K(t) g(\mathcal{L}) |$. We are actively working on refinements that would improve tightness under broader and more realistic training regimes.
>
>
> 3. Comparison to other lower bounds or convergence rates
>
> To the best of our knowledge, most existing convergence results (e.g., those based on the PL inequality or convexity assumptions) focus on upper bounds that characterize how quickly the loss decreases under specific regularity conditions. In contrast, our approach formulates necessary conditions via differential inequalities that constrain how rapidly the loss can decrease, irrespective of those assumptions. These perspectives are complementary: standard convergence rates offer best-case scenarios under assumptions, while our framework offers floor estimates and minimal training duration regardless of model convexity or smoothness. We believe combining both approaches could ultimately yield tighter, more informative constraints on resource usage for reaching a target loss. We plan to incorporate a brief comparative discussion in the revised version to better contextualize our results.
>
> While we did not investigate it, we believe it is possible to transfer those bounds to $d\theta/dt$ by expressing it as a function of  $d\mathcal{L}/dt$. We are open for any opportunity to work on the subject.

---

> > ### Comment · Reviewer_duBp · 2025-08-01
> > **Thank you for your rebuttal**
> >
> > Thank you for addressing my questions. Although I find your approach to establishing lower bounds interessting, the lack of clarity regarding tightness and effectiveness is the reason I’m standing by my current score.

---

### Official Review · Reviewer_C4e4 · 2025-06-30

**Clarity:** 1
**Significance:** 1
**Originality:** 2
**Rating:** 2
**Confidence:** 4

**Summary:**

The authors propose lower bounds for the value of the loss function during gradient descent training of parametric models in supervised learning. The bounds factor into two terms, one depending on the model and the dataset, while the other is determined by the loss function. The aim of the bounds is to estimate worst-case training time.

**Questions:**

It would benefit readability if the theoretical part is separated into propositions and their proofs, with the assumptions clearly stated.

Equation (29) does not describe adaptive optimizers, as the parameter update is in general not a linear function of the gradient.

How exactly is the minibatch loss floor plotted in Figure 5 calculated?

The argument of the function $g$ is inconsistent, being either $\mathcal{L}$, $\ell$ or $x$. Around equation (2), considering the data distribution over $\ell_i$ is a bit misleading. The learning rate is denoted $lr$ in equations, it would read better if it is only one letter. These are three of the number of ways notation could be improved.

**Ethical Concerns:**

["NO or VERY MINOR ethics concerns only"]

**Final Justification:**

The issues I had persist and the rebuttal raised another issue, hence I keep my rating as it was.

**Limitations:**

yes

**Quality:**

2

**Strengths And Weaknesses:**

1. Quality: The abstract claims that the bounds are tight, but Figure 5 demonstrates that they are in fact loose. Necessary convergence conditions refer to the proposed bound reaching a given threshold. The extension to adaptive optimizers in Section 8 seems to be wrong, as it assumes that the parameter update is a linear function of the gradient. The derivations seem to be correct modulo some typos. The claims are not collected into propositions. The last section clearly states that more work is needed for practical significance.

2. Clarity: Assumptions, claims and proofs are not separated, making the paper more difficult to follow. Notation is often unclear. It is not stated how the loss floor plotted in Figure 5 is calculated.

3. Significance: The proposed bounds are too loose to be significant. The idea that loss lower bounds can provide a way to estimate compute requirements could prove to be valuable if tight bounds are obtained.

4. Originality: Other works tend to focus on loss upper bounds, so that exploiting lower bounds seems to be novel. On the other hand, the proof techniques are less innovative.

---

> ### Author Rebuttal · Authors · 2025-07-27
>
> Thank you for your comments on the ideas presented in our submission. We appreciate the opportunity to clarify our assumptions, respond to your concerns, acknowledge current limitations of this work. We also regret that the current work did not meet expectations and hope the points below provide additional context and transparency.
>
>
> 1. On the structure and clarity of the paper
>
> We agree that the presentation would benefit from clearer separation between assumptions, claims, and proofs. In the current version, many derivations appear inline with the main text, which may have hindered readability. In hindsight, we underutilized the appendix, where most proofs and detailed derivations should have been placed. This was a trade-off to keep the narrative flowing, but we recognize that it compromised formal clarity. In future revisions, we intend to restructure the main theoretical results as explicit propositions or theorems, with assumptions clearly stated and technical proofs deferred to the appendix.
>
>
> 2. On extensions to other optimizers
>
> You're right to point out that the generalization proposed in Equations (29) and (30) was speculative. These expressions were meant to illustrate how the original SGD-specific flow might be adjusted when applied to momentum-based or adaptive optimizers, but we agree that this part lacks rigor. Since the submission, we have made a minor and yet limited progress with AdaGrad, showing that under an additional hypothesis, it fits in a different framework that has a different geometry. We plan to develop these results in a dedicated follow-up.
>
> 3. On the computation of loss floors in Figure 5
>
> Thank you for highlighting the need for clarification. As indicated in the text, we modeled upper bounds for the gradient norm as a product of a model-specific decay rate $K(t)$ (shorthand for $K(f, \theta(t), D)$ and a "shape" function $g(\mathcal{L})$ that only depends on the loss. Because $K(t)$ is intractable without assumption about the architecture of the model, we computed it from training logs, tracking the loss, gradient norms for each optimizer: SGD, AdaGrad, RMSProp, AdamW. These empirical values where then used as look-up table in an ODE solver to generate the numerical loss floor curves.  This retrospective procedure is not predictive, but it illustrates the behavior of the bounds given observed dynamics. Code for this process is openly available in the GitHub repository linked in the paper.
>
>
> 4. On notation inconsistencies
>
> We appreciate your remarks on the clarity of notation and agree there are inconsistencies that need to be addressed. Specifically:
>  - In Table 1, the function  g represents a shape function associated with each loss type. We used $x$ as a independent variable to emphasize functional form but acknowledge that this could be misleading since the letter $x$ was also used in the definition of the dataset D.
> - In section 4.2, a different function also denoted g was introduced in another context, It would have been clearer to use a distinct symbol.
> - The loss notation could also been clarified with a dedicated notation section. To summarize $\ell_i = \ell(y_i, f(x_i, \theta))$ is the per-sample loss, while $\mathcal{L}$ refers to the expected loss over a mini-batch.
> - The learning rate notations will be revised for conciseness and consistency
>
>
> We appreciate your thoughtful critique and hope this clarifies both our intent and the scope of the proposed work.

---

> > ### Comment · Reviewer_C4e4 · 2025-08-02
> >
> > Thank you for addressing my concerns. Regarding the computation of the loss floors, if the model-specific decay rate cannot be computed in advance, then even if the bounds were tight, I'm afraid they would not be very useful to anticipate convergence speed and to estimate minimum training time and energy requirements. As my other issues persist as well, I will keep my rating as it is.

---

### Official Review · Reviewer_A47x · 2025-07-02

**Clarity:** 2
**Significance:** 2
**Originality:** 2
**Rating:** 2
**Confidence:** 4

**Summary:**

The paper proposes general lower bounds for the time derivative of the loss under gradient flow for several different kinds of losses (MAE, MSE, categorical and multi-label binary cross-entropy). Extensions involving multi-component losses, regularization and SGD are also discussed. It is argued that the derived bounds can help estimate energy consumption during model training. A section with experiments shows learning curves of several optimization algorithms compared with the respective theoretical bounds.

**Questions:**

The paper needs a major revision. Key statements should be properly formulated, preferably as theorems with clear hypotheses and conclusions. Technical details of the proofs can be moved to the appendix. (Lack of) tightness of the bounds should be properly addressed. If optimization algorithms other than SGD are considered, they deserve separate careful analysis and discussion. The usefulness of the obtained bounds (e.g., for energy footprint estimation) should be demonstrated more convincingly.

**Ethical Concerns:**

["NO or VERY MINOR ethics concerns only"]

**Final Justification:**

The rebuttal has not changed my initial opinion and recommendations.

**Quality:**

2

**Strengths And Weaknesses:**

The merit of the paper is that it attempts to develop a general theory of loss lower bounds applicable to various losses and general problems. Unfortunately, I cannot  call this attempt successful.

While the paper is theoretical, it is written in a "stream of consciousness" style, as a chain of arguments and calculations without any theorem statements. To understand what the results are and under which assumptions they are obtained the reader basically needs to read attentively through the whole paper. Some assumptions are buried deep in the arguments (e.g., the bound for the categorical cross-entropy in Section 4.2 seems to be obtained under the assumption $Var(l)\\le 2^{-m} L^2$).

The derived bounds depend on the "instantaneous decay rate" $K=\\sqrt{\\tfrac{1}{n}\sum_{i=1}^n \\|\\nabla f(x_i,\\theta)\\|^2} $ which is unknown in general. As a result, the respective differential inequalities for the loss are not fully specified and hence do not generally produce concrete loss bounds. This significantly limits the usefulness of the obtained results.

As a way to overcome this difficulty, the paper considers linear models, since $K$ is known and constant in this case: $K=\\sqrt{\\tfrac{1}{n}\sum_{i=1}^n \\|x_i\\|^2}$. This is a very special case, and this is the only case in which we get a specific lower bounds for $L(t)$ (table 3). However, these bounds strongly underestimate the actual loss. In the MAE case, the bound $L(t)\\ge L_0-K^2t$ becomes negative and hence vacuous for sufficiently large $t$. In the MSE case, an explicit exact form of the loss is available:
$$L(t)=\sum_{m}c_me^{-2h_mt},$$
where $h_m\\ge 0$ are the eigenvaliues of the Hessian $H=\\tfrac{1}{n}\sum_{i=1}^n x_i x_i^T$ and $c_m\\ge 0$ are the respective initial eigencomponents. We can compare this exact solution with the bound $L(t)\\ge L_0e^{-2K^2 t}$ from table 3. We have $L_0=\sum_m c_m$ and $K^2=\\tfrac{1}{n}\sum_{i=1}^n \\|x_i\\|^2=Tr(H)=\\sum_m h_m,$ so that the bound $L(t)\\ge L_0e^{-2K^2 t}$ can be equivalently written as
$$L(t)\\ge\\Big(\sum_m c_m\\Big)e^{-2\\sum_m h_m t}.$$
We see that the proposed bound can substantially underestimate the actual loss.

Discussion of the energy footprint in section 7 is unspecific; no examples of actual numbers for the energy consumption are provided. Given the lack of tightness of the bounds, it is unclear how accurate and useful such results could be.

Section 8 with experiments is confusing. First, it provides results not only for SGD, but also for other algorithms (AdamW, etc.), not theoretically analyzed up to this point. The section contains a very brief sketch of ideas related to these extensions, but this discussion is too short and obscure. For example, it is not clear why the generalized equation (29) is first-order, since the natural continuous-time analog of dynamics with momentum is second-order. It is accordingly not clear what are the theoretical bounds shown in Figure 5, and why they hold. Next, it is puzzling that for RMSProp the bound is violated in the experiments. This suggests that the proposed bounds are neither tight (as both Figure 5 and my comments above show) nor reliable.

---

> ### Author Rebuttal · Authors · 2025-07-25
>
> Thank you for your detailed and constructive comments on the ideas that we proposed in the paper. Below we respond to key concerns, clarify assumptions and transparently acknowledge the limits of the present work.
>
>
> 1. Lack of structure & missing theorems
>
> We agree that the current proposal could benefit from more explicit theorem statements and clearer highlighting of assumptions. This was a trade-off decision to streamline exposition of ideas but recognize that this came at the cost of some formal clarity. In future revision we will restructure key results with proofs in the appendix instead of in the main paper and remove superfluous sections.
>
>
> 2. Dependence on Instantaneous rate K(t)
>
> We acknowledge that the SGD differential inequality inequality depend on the hardly tractable, architecture-dependent and dataset-dependent rate K(t) which limit its predictive power. But to demonstrate (in section 8,) that in practise a lower bound can be estimated, we computed empirical K(t) from training logs with SGD, AdaGrad, RMSProp and Adam.
>
> We later used them in look-up tables for numerical integrations. While it does not yield a closed-form, it enabled us to test how well the bound characterize loss dynamics retrospectively. As indicated in the paper, the code used is available in the GitHub repository indicated in the paper.
>
>
> 3. Other optimizers
>
> The primary focus of the paper was on SGD but we included those experiments with other optimizers to demonstrate both the promise and the limits of the proposed framework. We agree that the brief mention of Eq.29/30 was under-explained. We have since made some explorations to search for a more rigorous differential inequality for AdaGrad and the other optimization, so far with minor and limited success for AdaGrad.
>
> Regarding Figure 5, where RMSProp appears to violate the bound obtained with the SGD differential inequality reinforce the idea that each optimizer reauires its own treatment.
>
>
> 4. Limited Tightness of the Bounds
>
> Indeed the lowers bounds are not tight in general and in some cases (e.g MAE) could be vacuous. We appreciate this critique and want to pointed that the tightness is driven by coarse inequalities (e. triangle inequality, Cauchy-Schwarz), which are difficult to circumvent analytically.
>
> Arguably, an alternative but intractable approach to upper bound the norm of the loss gradient is by starting from
>
> $|| \nabla \mathcal{L} ||^2 =  \frac{1}{n^2}  \sum_{i=1}^{n} \sum_{j=1}^{n}  \nabla \ell_i \cdot \nabla \ell_j $
>
> Then problem boils down to taking into account the angles between the gradients for any pair of trainset samples. We are exploring techniques to take it into account.
>
> 5. The categorical cross-entropy assumption
>
> The upper bound in Section 4.2 depends on the assumption that $Var[\ell] \leq 2^{-m} E[\ell]^2$ which is non-rigorous and only heuristically motivated with the Bhatia-Davis inequality and numerical experimentations. Our was never to be misleading about this and we frame this as a plausible conjecture rather than a theorem. We can move this to appendix or label it as a heuristic if needed. With that said, the Tanh(x/2) results remains the closest conjecture-free upper bound.
>
>
> 6. Energy discussion
>
> We agree that the discussion in Section 7 could be better grounded with actual measurements. Our intent was to show that once K(t) is estimated or bounded, the integral bound on loss decay can serve as a proxy for energy or training duration comparison motived by the fact that it is necessary for that bound to pass below a arbitrary threshold before the training loss actually does too.
>
> 7. Summary:
> While we understand the concerns raised, we believe the paper contributes in:
> -  proposing another perspective on loss decay via gradient norms
> -  an empirical pipeline for retrospective estimation of decay rates K(t)
> -  initial theoretical and experimental evidence that such bounds can distinguish optimization regimes
>
> We appreciate the opportunity to improve the work and thank you again for your feedback.

---

> > ### Comment · Reviewer_A47x · 2025-08-04
> >
> > Thank you for this response and providing more context of your work. I keep my score.

---

### Decision · Program_Chairs · 2025-09-17

**Decision:**

Reject

**Comment:**

This paper analyzes the lower bound of the loss during gradient descent training, illustrating the irreducible loss floor.
It is also shown that this can be used to predict the worst-case training time and the corresponding energy footprint.

The paper approaches the problem from a theoretical perspective, but there are no formal assumptions, nor are there concrete theorem statements provided.
There are also concerns regarding the tightness of the bounds and the practical implications of the findings.
These issues require major revisions to strengthen the theoretical contributions and clarify the practical relevance of the results.